# The Rapid Upper Limb Assessment Among Traditional Krajood (*Lepironia articulata*) Handicraft Workers: A Case Study in Southern Thailand

Kaknokrat Chonsin [1] , Suthasini Buaphet [1], Jutamas Intarasombut [1], Aujchariya Chotikhun [2] and Jitralada Kittijaruwattana [2],*

[1] Faculty of Science and Technology, Suratthani Rajabhat University, Mueang, Surat Thani 84100, Thailand; kaknokrat.cho@sru.ac.th (K.C.); suthasini.bua@gmail.com (S.B.); jutamas254325@gmail.com (J.I.)

[2] Faculty of Science and Industrial Technology, Prince of Songkla University, Surat Thani Campus, Mueang, Surat Thani 84000, Thailand; aujchariya.c@psu.ac.th

* Correspondence: jitralada.k@psu.ac.th

**Abstract:** Musculoskeletal disorders (MSDs) are associated with awkward postures, causing health problems for workers. MSDs impact physical activity levels and decrease professional work capacity. The objective of this study is to investigate the ergonomic risks in a handicraft community enterprise group using Krajood as the main raw material. The sample group consisted of craftsmen who engage in woven bags, and it was selected using inclusion and exclusion criteria. Data were collected with a general information questionnaire, a risk assessment questionnaire for musculoskeletal disorders, and the Rapid Upper Limb Assessment (RULA) worksheet. The results indicate that musculoskeletal disorders were experienced by all the workers during the past year, with pain or discomfort in all 12 body parts. Moreover, most commonly, the pains were in the shoulders, upper back, lower back, and hands/wrists on both the left and the right side. The lower back exhibited a 100% prevalence of symptoms. The risk assessment by RULA indicated that the jobs had the highest possible total risk score at 7 points (45%), which needs to improve immediately. The top three high-risk work processes were the product line hammering steps, using a sewing machine to form the product, and the weaving and forming stage. Therefore, this study provides critical information for the craftsmen and their employers to improve workers' health and production efficiency.

**Keywords:** musculoskeletal disorders; risk assessment; Rapid Upper Limb Assessment; Krajood

## 1. Introduction

Musculoskeletal disorders (MSDs) are defined as health problems in the system consisting of bones, joints, skeletal muscles, tendons, and motor nerves, including all forms of ill health ranging from light, transitory disorders to irreversible, disabling injuries [1]. Musculoskeletal disorders stemming from work are the most important type of occupational health problems in both developing and developed countries. They amount to a record-breaking problem worldwide and cause huge losses each year. Work-related musculoskeletal disorders (WMSDs) are associated with lifting heavy loads, other forceful exertions, awkward posture, repetition, and whole-body vibration [2].

In 2019, globally, 2.41 billion individuals had conditions that would benefit from rehabilitation, contributing to 235 to 392 million years of life lived with disability (YLDs). Approximately 1.71 billion people worldwide live with musculoskeletal conditions, including lower back pain, neck pain, fractures, or other injuries, including osteoarthritis,

amputation, and rheumatoid arthritis [3]. The European Agency for Safety and Health at Work (EASHW) reported that MSDs are the most common occupational disease type among the recognized occupational diseases, accounting for 59% of the total [4]. The most common professions associated with diseases are musculoskeletal disorders, such as joint pain, back pain, and muscle pain. Those problems also increase production, medical, and insurance costs, while degrading work capacity and causing loss of employees to disabilities [5].

Basketry is a skilled occupation using art, craft, and ancient wisdom in making interwoven objects, usually containers made from natural flexible materials such as twigs, grass, osiers, bamboo, and rushes, while the modern variants can be made from plastics or other synthetic materials. The goods are entirely handcrafted, which has gradually become a context for luxury items, and this generates value-added products from natural materials. Originally, these products were local handicraft products made by splitting, knitting, and weaving [6]. In the southern peninsular part of Thailand, traditional basketry has used sedge as the raw material [7].

*Lepironia articulata* is a weed, also known as sedge, a tall, slender, tightly rhizomatous macrophyte that forms large dense swords of foliage belonging to the family Cyperaceae. It is found in Madagascar, India, Sri Lanka, southern China, and generally in Southeast Asia, and it grows in all types of soils associated with wetlands, including poor soils [8]. In extensive open swamp areas of Thailand and Malaysia, *Lepironia articulata* stands out as one of the key primary plants. This species has been introduced to groundwater-dependent wetland ecosystems as it can improve the quality of graywater in the swamp and reduce many types of contaminants, and it reduces greenhouse gasses by carbon capture to stocks in the products [9–11]. Despite the obvious importance of the species to wetland ecosystems, the grass can grow up to 2 m tall and live above water [12]. It is also commonly called gray sedge or Krajood in Thai and can be used in sustainable woven products such as baskets, hats, and mats. The basketry generates extra income for local families in southern Thailand [7].

Basketry combines artistic expression with practical functionality in handmade items. The craftsman has to have experience and master the necessary techniques. The posture and motions of the workers are important for productivity and also determine the risk of MSDs in the workplace. Therefore, a study of ergonomics can improve the worker's performance while reducing stress and fatigue at work. The application of ergonomics is extremely significant for manual activities, directly affecting an employee's physical and mental health [13,14].

Rapid Upper Limb Assessment (RULA) was established to assess children's posture, especially when using computers at school [15–17]. It is a subjective observation method for posture analysis that focuses on the upper body and includes the lower body [18]. This method is still commonly used because it is convenient, low-cost, and more flexible than other procedures used in field studies [19–21].

Therefore, this study focused on performing an ergonomic risk assessment among craftsmen by using the RULA technique. Appropriate postures and activities should improve productivity and reduce musculoskeletal disorders in workers who make shoulder bags from Krajood as the raw material in Thailand. Moreover, it can provide guidelines for workstation improvements or suggest accessories that help prevent further injuries, thereby decreasing musculoskeletal disorders in the handicraft community.

## 2. Materials and Methods

The assessment of ergonomic risks concentrated on the Ban Huai Luek Krajood handicraft community enterprise in the Phunphin district of Surat Thani province, Thailand. This community has a century-long tradition of practicing the traditional arts and crafts

of basketry. The production of shoulder bags from *Lepironia articulata* as the raw material was examined. The sample comprised 20 voluntary participants who met specific criteria. The criteria required that the participants must be over 18 years old, speak Thai, have over one year of experience, not be pregnant, and be volunteering for the study. All participants received an explanation of the information related to this research. This research has sought and was granted ethical approval. All of the data were stored in password-protected computer files.

The cross-sectional study employed a general questionnaire in the Thai language, the risk of musculoskeletal disorder questionnaire using the Standard Questionnaire of Division of Occupational and Environmental Diseases, Department of Disease Control, Ministry of Public Health, Thailand, adapted from the Nordic Musculoskeletal Questionnaire (NMQ) [22,23], and the Rapid Upper Limb Assessment (RULA) established by Lynn McAtamney and Nigel Corlett, 1993 [24].

The standard questionnaire of the Division of Occupational and Environmental Diseases had six parts: (1) general information, (2) health conditions that are associated with risk of musculoskeletal injury, (3) work history/hobbies, (4) symptom survey of musculoskeletal disorders, (5) work environment assessment, and (6) causes of musculoskeletal illness. Specialists evaluated all the questionnaires. After considering the opinions of three experts who are academic professionals in occupational health and safety for at least 5 years regarding the questionnaire's validity, the index of item-objective congruence (IOC) was calculated using Formula (1) [25] to the value 0.92, indicating validity for this study.

The index of item-objective congruence (IOC) used the following scoring criteria: "Sure" there is consistency has a score level of 1, "Not sure" there is consistency has a score level of 0, and "Sure" there is no consistency has a score level of $-1$. Following that, the experts' scores were used to calculate the IOC as follows:

$$\text{IOC} = \frac{\sum R}{N} \tag{1}$$

where IOC is the index of item-objective congruence, $\sum R$ is the sum of expert opinion scores, and N is the number of experts.

The reliability coefficient of the questionnaire was assessed by using 30 volunteers from a sample group of rubber farmers in Surat Thani. The data collected were analyzed for Cronbach's alpha, calculated as follows:

$$\alpha = \frac{k}{k-1}\left(1 - \frac{k\sum s_i^2}{s_t^2}\right) \tag{2}$$

where $\alpha$ is the coefficient of reliability, $\sum S_i^2$ is the variance of the score, K is the total number of questions, and $St^2$ is the variance of sum scores.

The reliability categories based on Cronbach's alpha are: as follows: (1) $\alpha \geq 0.9$ is excellent, (2) $0.9 \geq \alpha \geq 0.8$ is good, (3) $0.8 \geq \alpha \geq 0.7$ is acceptable, (4) $0.7 \geq \alpha \geq 0.6$ is questionable, (5) $0.6 \geq \alpha \geq 0.5$ is poor, and (6) $0.5 \geq \alpha$ is unacceptable [26]. The coefficient of reliability found was 0.78, being adequate and acceptable in overall quality for this study.

A body map in the Musculoskeletal disorder questionnaire indicated the twelve symptom sites neck, shoulders, upper back, lower back, upper arms, lower arms, elbows, wrists/hands, hips/thighs, knees, calves, and ankles/feet. Respondents were examined to see if they have had any musculoskeletal trouble in the past year, preventing normal activity [23].

VDO was recorded with an iPhone 12, Apple, China, with HDR of Dolby Vision in 4K at 30 fps. The three cameras were in the front, on the left, and on the right, and there were several sessions for approximately 5 min once they were worked on a task. The VDO files

were checked flame by flame and paused for static investigation to identify awkward body postures and measure angles for the RULA scores.

A common statistic used to assess agreement is the Intraclass Correlation Coefficient (ICC). Intra-rater Reliability was used for agreement measurement based on this study. One-way random effects, absolute agreement, and single rater/measurement model were evaluated [27]. The ICC was 0.932, which indicates good communication among researchers and participants, calculated as follows:

$$\text{ICC} \, (1,1) = \frac{\text{MS}_R - \text{MS}_W}{\text{MS}_R + (k+1)\text{MS}_W} \tag{3}$$

where ICC is the Intraclass Correlation Coefficient, in (1,1) the first number refers to model 1 and the second indicates a single rater/measurement, $\text{MS}_R$ is the mean square for rows, $\text{MS}_W$ is the mean square for residual sources of variance, and k is the number of raters/measurements.

The RULA had 15 steps in a survey, using the RULA employee assessment worksheet [24]. The data were collected by video recording workers at each workstation to investigate their postures slowly thereafter [28]. The interpretations of the final score are as follows: 1 or 2 means "acceptable", 3 or 4 means "investigate further", 5 or 6 means "investigate further and change soon", and 7 means "investigate and change immediately". The postures were collected by observing the behaviors of several workers. The three cameras were on the front, left, and right in several recording sessions. Each workstation was observed using a video recorder, and the videos were meticulously assessed frame by frame. The process of making shoulder bags and postures is summarized in Figure 1. The data were collected and analyzed by using descriptive statistics in Microsoft Excel 365®.

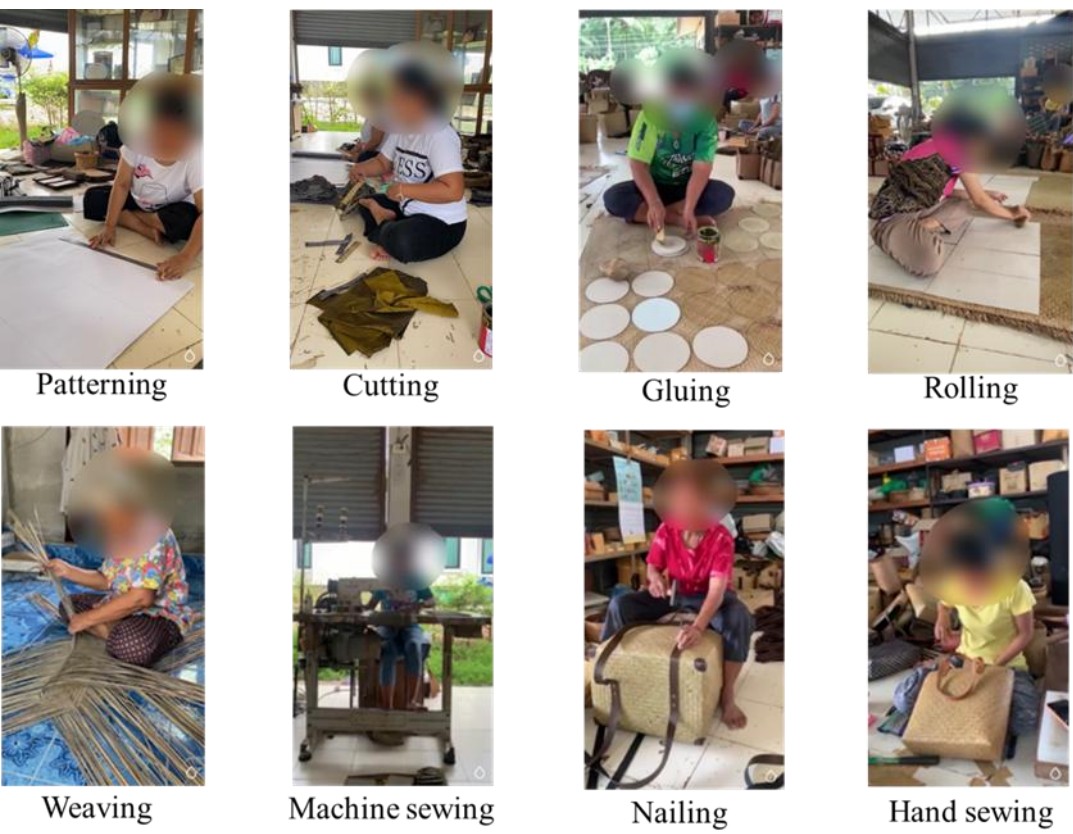

**Figure 1.** The stage processing and postures of shoulder bag production.

## 3. Results

Figure 1 illustrates the postures of workers involved in the manufacture of shoulder bags, revealing that the nailing and weaving processes account for approximately 25% of the overall workload. Figure 2 further displays the percentage of workers engaged in these production activities namely nailing (25%), cutting (5%), gluing (20%), patterning (5%), machine Sewing (10%), weaving (25%), hand sewing (5%), and rolling (5%). Notably, the weaving requires a comparatively higher level of experience and skill, and it is predominantly performed by women aged 30 and older. This trend reflects the demographics of artisans in this industry, as men in this community typically are engaged in agricultural activities.

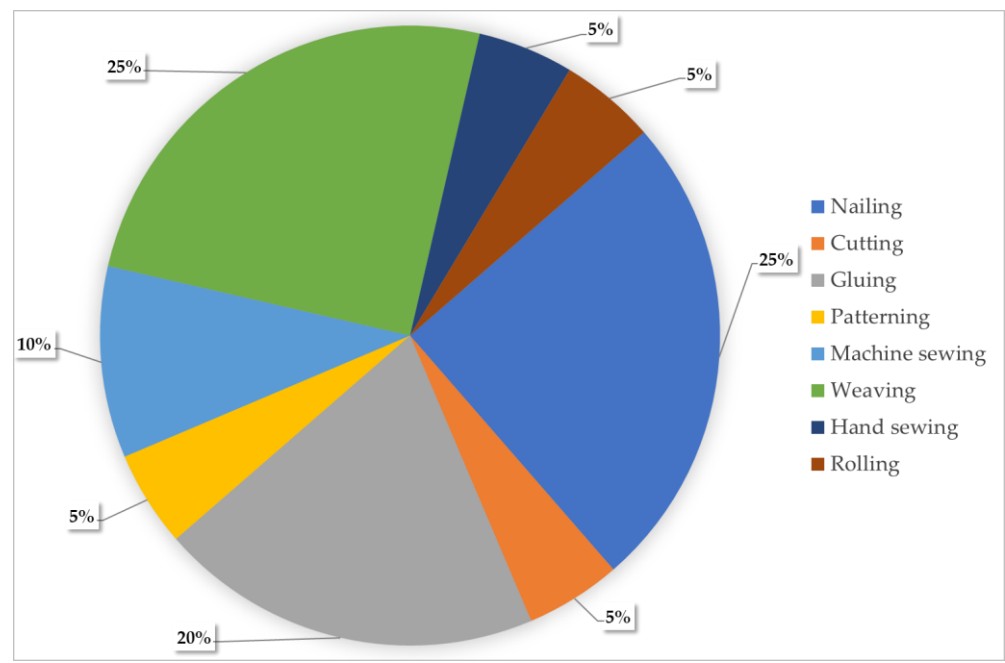

**Figure 2.** Percentages of workers in the production stages of shoulder bags.

Table 1 provides general information regarding health conditions associated with the risk of musculoskeletal injuries among the participants. The study found that 100% of the participants were female, with 80% reporting no significant health issues; however, 90% had experienced a serious accident in the past. Additionally, all participants were non-smokers, and 75% abstained from alcohol consumption. Despite this, health conditions such as hypertension, obesity, and diabetes mellitus were noted within the basketry community examined in this study.

Table 2 summarizes the results from using a symptom survey of musculoskeletal disorders among the basketry workers. Thirteen or 65% of the participants reported sometimes feeling tired after work, while 10% of them always had this problem. In any single year, the craftsmen always felt discomfort with the body during work. The most abnormal symptoms were reported as pain in the lower back, upper back, wrists/hands, upper arms, and right forearm, in rank order. Indeed, such symptoms have often been suffering for 5 years. Previous studies using the NMQ survey have reported on perceptions of various work-related risk factors and musculoskeletal symptoms [29–32].

**Table 1.** Health conditions associated with the risk for musculoskeletal injury in the study subjects.

| Health Condition | Participant (*n*) | Percentage (%) |
|---|---|---|
| Health issues | | |
| None | 16 | 80 |
| Medical history | 4 | 20 |
| - Obesity | 1 * | 5 ** |
| - Hypertension | 4 * | 20 ** |
| - Diabetes Mellitus | 1 * | 5 ** |
| Ever had a serious accident | | |
| Ever | 18 | 90 |
| Never | 2 | 10 |
| Smoking | | |
| Smoker | 0 | 0 |
| Non-smoker | 20 | 100 |
| Drinking | | |
| Every day | 0 | 0 |
| Sometime | 5 | 25 |
| Never | 15 | 75 |

* One of the participants reported more than one health issue. ** Percentage is based on the total number of participants.

**Table 2.** Symptom survey of musculoskeletal disorders among basketry workers.

| Issue | Participant (*n*) | Percentage (%) |
|---|---|---|
| Feeling tired after work | | |
| Never | 0 | 0 |
| Sometime | 13 | 65 |
| Often | 5 | 25 |
| Always | 2 | 10 |
| Has any musculoskeletal trouble occurred in the last 12 months? | | |
| Yes | 20 | 100 |
| No | 0 | 0 |
| In a single year felt discomfort in body | | |
| Ever | 20 | 100 |
| Never | 0 | 0 |
| Body region with abnormal symptoms * | | |
| Lower back | 8 | 40 |
| Upper back | 5 | 25 |
| Wrists/hands | 4 | 20 |
| Upper arms | 2 | 10 |
| Right forearm | 2 | 10 |
| How long have the abnormal symptoms lasted? | | |
| 1 year | 4 | 20 |
| 2–3 years | 8 | 40 |
| 4–5 years | 6 | 30 |
| >5 years | 2 | 10 |

**Table 2.** *Cont.*

| Issue | Participant (*n*) | Percentage (%) |
|---|---|---|
| In the past year, how many times such abnormal symptoms have occurred? | | |
|     1–5 times | 15 | 75 |
|     6–10 times | 4 | 20 |
|     >10 times | 1 | 5 |
| What is causing this? | | |
|     Sitting at work for a long time | 12 | 60 |
|     Using hands to work | 2 | 10 |
|     Sitting on the floor | 1 | 5 |
|     Tightening arms while working | 5 | 25 |

* One of the participants reported more than a single health issue.

The severity of symptoms across various body regions was examined, and the findings are summarized in Table 3. Twelve body parts were assessed for the craftsmen, namely neck, shoulders, upper back, lower back, upper arms, lower arms, elbows, wrists/hands, hips/thighs, knees, calves, and ankles/feet. Each body part was analyzed separately for the left side, right side, or both sides. The analysis revealed no significant differences in symptom severity among these body regions for participants experiencing severe symptoms. Table 4 presents the total risk scores and their interpretations according to the Rapid Upper Limb Assessment (RULA), based on work postures observed at each stage.

**Table 3.** The severity of symptoms by body region.

| Symptom and Its Severity Score | Left | | Right | | Both side | |
|---|---|---|---|---|---|---|
| | Participant (*n*) | Percentage (%) | Participant (*n*) | Percentage (%) | Participant (*n*) | Percentage (%) |
| 1. Neck | | | | | | |
| - 0 score | 4 | 20 | 4 | 20 | 4 | 20 |
| - 1 score | 16 | 80 | 16 | 80 | 16 | 80 |
| 2. Shoulders | | | | | | |
| - 0 score | 0 | 0 | 0 | 0 | 0 | 0 |
| - 1 score | 20 | 100 | 20 | 100 | 20 | 100 |
| 3. Upper back | | | | | | |
| - 0 score | 0 | 0 | 0 | 0 | 0 | 0 |
| - 1 score | 20 | 100 | 20 | 100 | 20 | 100 |
| 4. Lower back | | | | | | |
| - 0 score | 0 | 0 | 0 | 0 | 0 | 0 |
| - 1 score | 20 | 100 | 20 | 100 | 20 | 100 |
| 5. Upper arms | | | | | | |
| - 0 score | 1 | 5 | 1 | 5 | 1 | 5 |
| - 1 score | 19 | 95 | 19 | 95 | 19 | 95 |
| 6. Lower arms | | | | | | |
| - 0 score | 1 | 5 | 1 | 5 | 1 | 5 |
| - 1 score | 19 | 95 | 19 | 95 | 19 | 95 |
| 7. Elbows | | | | | | |
| - 0 score | 10 | 50 | 10 | 50 | 10 | 50 |
| - 1 score | 10 | 50 | 10 | 50 | 10 | 50 |
| 8. Wrists/hands | | | | | | |
| - 0 score | 0 | 0 | 0 | 0 | 0 | 0 |
| - 1 score | 20 | 100 | 20 | 100 | 20 | 100 |
| 9. Hips/thighs | | | | | | |
| - 0 score | 7 | 35 | 7 | 35 | 7 | 35 |
| - 1 score | 13 | 65 | 13 | 65 | 13 | 65 |
| 10. Knees | | | | | | |
| - 0 score | 5 | 25 | 5 | 25 | 5 | 25 |
| - 1 score | 15 | 75 | 15 | 75 | 15 | 75 |
| 11. Calves | | | | | | |
| - 0 score | 8 | 40 | 8 | 40 | 8 | 40 |
| - 1 score | 12 | 60 | 12 | 60 | 12 | 60 |
| 12. Ankles/feet | | | | | | |
| - 0 score | 11 | 55 | 11 | 55 | 11 | 55 |
| - 1 score | 9 | 45 | 9 | 45 | 9 | 45 |

**Table 4.** Total RULA risk scores by worker and work stage, based on work postures.

| Worker | RULA Score | | | | | | | |
|---|---|---|---|---|---|---|---|---|
| | Nailing | Cutting | Gluing | Patterning | Machine Sewing | Weaving | Hand Sewing | Rolling |
| 1 | 6 | | | | | | | |
| 2 | | 5 | | | | | | |
| 3 | | | 3 | | | | | |
| 4 | | | | 7 | | | | |
| 5 | | | 7 | | | | | |
| 6 | | | 6 | | | | | |
| 7 | | | | | 6 | | | |
| 8 | 7 | | | | | | | |
| 9 | 5 | | | | | | | |
| 10 | 7 | | | | | | | |
| 11 | | | | | | 5 | | |
| 12 | | | | | | 4 | | |
| 13 | | | | | 6 | | | |
| 14 | | | | | | 6 | | |
| 15 | | | | | | 3 | | |
| 16 | | | | | | 7 | | |
| 17 | | | | | | | 7 | |
| 18 | 7 | | | | | | | |
| 19 | | | 7 | | | | | |
| 20 | | | | | | | | 7 |

Table 5 demonstrates an analysis of the different working postures of workers. Six of the eight postures of activities are sitting on the floor with a hard surface. Their postures are sitting on the floor, sitting on the chair, back bent forward and twisted, both arms below shoulder level, sitting with both knees bent, and sitting with legs crossed. The stick diagram provided the information by collecting and analyzing from the video examinations of 20 participants.

**Table 5.** Analysis of different working postures of the basketry workers.

| Activities | Postures | Stick Diagram |
|---|---|---|
| Nailing | Sitting on the chair, back bent forward and twisted, both arms below shoulder level, holding right hand with a hammer, sitting with both knees bent, a weight of 10 kg or less |  |
| Cutting | Sitting on the floor, back bent forward and twisted, both arms below shoulder level, sitting with legs crossed, a weight of 10 kg or less |  |
| Gluing | Sitting on the floor, back bent forward and twisted, both arms below shoulder level, sitting with legs crossed, a weight of 10 kg or less |  |

**Table 5.** *Cont.*

| Activities | Postures | Stick Diagram |
|---|---|---|
| Patterning | Sitting on the floor, back bent forward and twisted, both arms below shoulder level, sitting with legs crossed, a weight of 10 kg or less | |
| Machine Sewing | Sitting on the chair, back bent forward and twisted, both arms below shoulder level, sitting with both knees bent, a weight of 10 kg or less | |
| Weaving | Sitting on the floor, back bent forward and twisted, both arms below shoulder level, sitting with legs crossed, a weight of 10 kg or less | |
| Hand sewing | Sitting on the floor paddling, back bent forward and twisted, both arms below shoulder level, sitting with legs crossed, a weight of 10 kg or less | |
| Rolling | Sitting on the floor, back bent forward and twisted, both arms below shoulder level, sitting with legs crossed, a weight of 10 kg or less | |

The results of this study indicate that those workers exhibiting the lowest overall risk scores, namely from 3 to 4 points, were only 10% of the sample. This suggests the need for further research and ongoing monitoring to develop new work practices (see Table 6). A total risk score of 5 or 6 signifies that the task may pose emerging problems; therefore, additional studies should be conducted, and conditions should be improved for 45% (*n* = 9) of the workers. The highest total risk score category, with a score of 7 or higher, indicates significant ergonomic issues that require immediate intervention, and this was assigned to 45% (*n* = 9) of the workers. The three work stages identified as having the highest overall risks were nailing, sewing, and weaving.

**Table 6.** Risk level classification by RULA.

| Score Range | Interpretation | Participant (*n*) | Prevalence (%) |
|---|---|---|---|
| 1 or 2 | This is acceptable but there may be some ergonomic problems with repeated operations. | 0 | 0 |
| 3 or 4 | Further studies are required and monitoring for new design work. | 2 | 10 |
| 5 or 6 | The job was starting to get problematic. More studies should be required and should be improved. | 9 | 45 |
| 7 or more | The job had an ergonomic problem and must be updated immediately. | 9 | 45 |

## 4. Discussion

The findings of the current study provide an assessment of the work conditions in a specific case of handicraft work. Typically, sewing machine workers can experience musculoskeletal symptoms and associated risk factors [33]. The high frequency of neck/shoulder and back complaints obtained in the present study is also in agreement with previous studies among relatively similar occupations, namely those involving sewing processes [34,35]. In this study, musculoskeletal pain and discomfort were major problems associated with the nailing stage. The craftsmen have to do the job all day for 8 h. The neck and body of workers are slightly bent to grab the straps up to measure them before nailing the strap holes. They use the right hand to hold the hammer to drive the nail, requiring some force to apply the hammer. Then, the left hand is holding the nail when the hammer hits its head, and the left hand needs to be tensed to make bag strap holes, continually all day long. The results indicate that most workers had backaches, especially in the lower back. Most of them sit with their back bent, hunching over and leaning forwards, which contributes to fatigue and risks to the musculoskeletal system. However, the limitations of this study were due to the small population of Krajood craftsmen, which induced a small sample size, and the designs of handbags were not being such that only one design could be assigned for one worker. If this community had more demand in their business for green products, a future study on an expanded sample of craftsmen might be possible.

RULA assessments are still an approach to assessing the risks of musculoskeletal disorders in real-life situations, and they are useful as part of an ergonomic assessment in various workplaces [36]. Ergonomic risk assessment of Ban Huai Luek Krajood handicraft community enterprise indicated that RULA is a reasonably reliable tool when applied to traditional basketry. It is important to consider the current findings in the context of cross-sectional data collection with RULA as an ergonomics risk assessment [37]. However, it is also necessary to highlight that the data depends on self-reporting, whose reliability and accuracy are subject to recall [38].

However, this study assessed a real-life situation among workers who have, on average, 3 years of experience and at least one year in their workstation. This was adopted from the method used by McAtamney and Corlett (1993) [24] to establish the reliability of the use of RULA with adults. A present review reported that workers who perform daily activities can suffer from musculoskeletal disorders, such as those that could be addressed by healthcare professionals using RULA ergonomic tools [39]. In this study, most of the basketry workers indicated having entire bodies on both the left and the right sides experience fatigue. Awkward postures occur because the work stages require sitting for a long time, with repetitive motions, and unneutral posture. Some participants stated that their existing musculoskeletal illnesses were related to their jobs. Therefore, our findings suggest that the handicraft community should have standard operating procedures (SOPs)

and work schedules developed for each workstation. The appropriate postures and time break plan can reduce MSDs and improve work efficiency [40–42]. In addition, the training for each workspace should be explained to all workers for improving occupational health and safety in manufacturing settings [43].

This study, while insightful, is constrained by a small sample size due to the limited population of Krajood craftsmen in the studied community. This limitation underscores the unique value of the findings, as they provide a detailed glimpse into the ergonomics of a specialized traditional craft that might otherwise be underrepresented in broader occupational health studies. Expanding this research to include more craftsmen and other handicraft communities in future studies could offer comparative insights and validate the patterns observed. Additionally, the diversity of handbag designs led to variations in tasks assigned to individual workers. While this variability may influence task-specific risk assessments, it also reflects the community's authentic working conditions, enhancing our findings' real-world applicability. Future research could benefit from controlling design variability or exploring how different product designs impact ergonomic risks.

## 5. Conclusions

The results of this study demonstrate that basketry workers had high-risk scores, reaching up to seven. This means that the work processes have ergonomic problems and must be reorganized immediately. The top-ranked three stages in shoulder bag manufacture were nailing, sewing, and weaving of the products, in decreasing order of risks. The participants thought that working posture affects joint and muscle pains during the work environment assessment. Furthermore, the workers held the belief that their current musculoskeletal ailments were linked to their occupations, which can be evaluated by NMQ. It is imperative to examine workstations, and prompt adjustments are essential for handicraft work and craftsmen possessing specialized skills. As a result, this study offers recommendations and facilitates targeted improvements for high-priority, high-risk workstations. These suggestions propose changes that are necessary to either the workstation itself or the organization of work, aiming to alleviate the musculoskeletal disorders experienced by artisans based on the study's findings. Additionally, it establishes guidelines to ensure proper postures in basketry work should be pursued. Following the transformation of the workplace and workers' behaviors, this study expresses confidence in the enduring continuity of this profession across generations.

**Author Contributions:** Conceptualization, K.C., S.B. and J.I.; data curation, J.K.; formal analysis, S.B., K.C. and J.K.; investigation, S.B., J.I. and K.C.; methodology, K.C. and J.K.; project administration, J.K.; Resources, S.B. and J.I.; supervision, K.C. and A.C.; writing—original draft, K.C., J.K. and A.C.; writing—review and editing, J.K. and A.C. All authors have read and agreed to the published version of the manuscript.

**Funding:** This research received no external funding.

**Institutional Review Board Statement:** The study was conducted in accordance with the Declaration of Helsinki, and the protocol was approved by the Ethics Committee of Human Research Ethics Committee, Suratthani Rajabhat University (SRU-EC2022/023) on 25 February 2022.

**Informed Consent Statement:** Informed consent was obtained from all subjects involved in the study.

**Data Availability Statement:** The data presented in this study are available on request from the corresponding author.

**Acknowledgments:** This work was partially supported by a grant from Suratthani Rajabhat University and sponsored by the government budget allocated to Prince of Songkla University. The authors express gratitude to Seppo Juhani Karrila for his invaluable contributions in scientific language proofread, revision, and editing.

**Conflicts of Interest:** The authors declare that there is no conflict of interest regarding the publication of this paper.

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
