# Peer review of "The Rapid Upper Limb Assessment Among Traditional Krajood (Lepironia articulata) Handicraft Workers: A Case Study in Southern Thailand"

_applsci, doi:10.3390/app15063142_

Round 1
Reviewer 1 Report
Comments and Suggestions for Authors
Thank you for the opportunity to review this manuscript, which considers some interesting, applied issues.
Thank you to the authors for the effort they put into creating the paper “Musculoskeletal Disorders Risk Assessment among Traditional Krajood (Lepironia articulata) Handicraft Workers: A Case Study in Southern Thailand”.
My biggest concern regarding this manuscript is the extremely small sample size. Due to the aforementioned fact, it is impossible to draw adequate conclusions on the topic of this paper. In the following, based on what I read in the manuscript, I believe that the process of selecting respondents for this study was not carried out adequately, in order to provide real facts.
It should be borne in mind that this study does not have an ethical committee decision and approval to conduct the research.
Also, the results are presented only in percentages. Therefore, this data also does not give a clear picture regarding Musculoskeletal Disorders Risk.
Due to all of the above, I believe that this manuscript in its current form is not adequate for further process in this Journal. If the authors set the topic differently, if they receive certain decisions, I believe that this can be an interesting topic for readers.
The last thing I want to mention is the large number of references that are older than 5 years. Authors should try to use newer literature that will provide readers with newer and more relevant information on the topic being covered.
Author Response
1.Comments and Suggestions for Authors
Thank you to the authors for the effort they put into creating the paper “Musculoskeletal Disorders Risk Assessment among Traditional Krajood (Lepironia articulata) Handicraft Workers: A Case Study in Southern Thailand”.
My biggest concern regarding this manuscript is the extremely small sample size. Due to the aforementioned fact, it is impossible to draw adequate conclusions on the topic of this paper. In the following, based on what I read in the manuscript, I believe that the process of selecting respondents for this study was not carried out adequately, in order to provide real facts.
It should be borne in mind that this study does not have an ethical committee decision and approval to conduct the research.
-The study was conducted in accordance with the Declaration of Helsinki, and the protocol was approved by the Ethics Committee of Human Research Ethics Committee, Suratthani Rajabhat University (SRU-EC2022/023) on February 25, 2022.
Also, the results are presented only in percentages. Therefore, this data also does not give a clear picture regarding Musculoskeletal Disorders Risk.
-I added more information in the new Table 5.
Due to all of the above, I believe that this manuscript in its current form is not adequate for further process in this Journal. If the authors set the topic differently, if they receive certain decisions, I believe that this can be an interesting topic for readers.
-The title of the manuscript was changed to “The Rapid Upper Limb Assessment among Traditional Krajood (Lepironia articulata) Handicraft Workers: A Case Study in Southern Thailand”.
The last thing I want to mention is the large number of references that are older than 5 years. Authors should try to use newer literature that will provide readers with newer and more relevant information on the topic being covered.
-The manuscript was revised, the references were added and updated.
Reviewer 2 Report
Comments and Suggestions for Authors
The introduction indicates the importance and prevalence of work-related musculoskeletal disorders and the possibility of using the RULA method to assess the ergonomic risk associated with work involving mainly the upper body. However, the purpose of the study was not defined. If the purpose of the study was to conduct it, then this purpose was achieved. However, in the introduction, the authors suggest that this study may provide guidelines for workstation improvements or suggest accessories that help prevent further injuries, thereby decreasing musculoskeletal disorders in the handicraft community. Unfortunately, no such guidelines were formulated.
I have no reservations about the way in which the group of respondents and the research methods were described.
There are some inconsistencies in the description of the study results. The title of Table 2 suggests that it contains a summary of results from using a symptom survey of musculoskeletal disorders among the basketry workers. It includes, among others: data concerning the body region with abnormal symptoms (lower back – 8 persons, upper back – 5 persons, wrists/hands – 5 persons, upper arms – 2 persons, right forearm – 2 persons). The severity of symptoms across various body regions was examined, and the findings are summarized in Table 3. The data presented therein show that 16 persons reported neck symptoms, shoulders – 20 persons, upper back – 20 persons, lower back – 20 persons, upper arms – 19 persons, lower arms – 19 persons, elbows – 10 persons, wirsts/hands – 20 persons, hips/thighs – 13 persons. No comment was made on the reasons for the differences in the information contained in both tables.
The assessment of the body position during work using the RULA method was made separately for each of the activities performed during the production of baskets. The presented data show that the most strenuous activities are patterning, hand sewing, rolling, slightly less strenuous are nailing, machine sewing, gluing, and the least strenuous are cutting and weaving. The authors rightly indicate that these assessments should be verified in studies involving a larger number of craftsmen. However, it can already be suggested that improvements should concern not only standard operating procedures (SOP) developed for each workstation, but also work schedules (possibility of rotation between positions).
To sum up, in my opinion the article is valuable, it shows how ergonomics can and should be applied also in research and evaluation of unique craft activities. Therefore, it should be published, but after clarifying all the doubts presented above.
Author Response
The introduction indicates the importance and prevalence of work-related musculoskeletal disorders and the possibility of using the RULA method to assess the ergonomic risk associated with work involving mainly the upper body. However, the purpose of the study was not defined. If the purpose of the study was to conduct it, then this purpose was achieved. However, in the introduction, the authors suggest that this study may provide guidelines for workstation improvements or suggest accessories that help prevent further injuries, thereby decreasing musculoskeletal disorders in the handicraft community. Unfortunately, no such guidelines were formulated.
-The manuscript was revised, and more detailed information was provided in the results and discussion section.
I have no reservations about the way in which the group of respondents and the research methods were described.
-The information was added to the methodology section.
There are some inconsistencies in the description of the study results. The title of Table 2 suggests that it contains a summary of results from using a symptom survey of musculoskeletal disorders among the basketry workers. It includes, among others: data concerning the body region with abnormal symptoms (lower back – 8 persons, upper back – 5 persons, wrists/hands – 5 persons, upper arms – 2 persons, right forearm – 2 persons). The severity of symptoms across various body regions was examined, and the findings are summarized in Table 3. The data presented therein show that 16 persons reported neck symptoms, shoulders – 20 persons, upper back – 20 persons, lower back – 20 persons, upper arms – 19 persons, lower arms – 19 persons, elbows – 10 persons, wirsts/hands – 20 persons, hips/thighs – 13 persons. No comment was made on the reasons for the differences in the information contained in both tables.
-Thank you for your comment. We have double-checked all information and add more information in the result section.
The assessment of the body position during work using the RULA method was made separately for each of the activities performed during the production of baskets. The presented data show that the most strenuous activities are patterning, hand sewing, rolling, slightly less strenuous are nailing, machine sewing, gluing, and the least strenuous are cutting and weaving. The authors rightly indicate that these assessments should be verified in studies involving a larger number of craftsmen. However, it can already be suggested that improvements should concern not only standard operating procedures (SOP) developed for each workstation, but also work schedules (possibility of rotation between positions).
-The manuscript was revised as your comment in the discussion.
To sum up, in my opinion the article is valuable, it shows how ergonomics can and should be applied also in research and evaluation of unique craft activities. Therefore, it should be published, but after clarifying all the doubts presented above.
Thank you very much for your kind support.
Reviewer 3 Report
Comments and Suggestions for Authors
This study provides a meaningful contribution to understanding ergonomic risks among traditional Krajood handicraft workers in Southern Thailand. The focus on musculoskeletal disorders (MSDs) in an under-researched occupational group is commendable. Below are key strengths, weaknesses, and suggestions for improvement:
Strengths
- Relevance: Addresses a critical occupational health issue in a traditional craft, highlighting the need for ergonomic interventions in artisanal sectors.
- Methodology: Appropriate use of RULA and Nordic Musculoskeletal Questionnaire (NMQ), combined with video analysis, strengthens the validity of findings.
- Practical Implications: Clear identification of high-risk work stages (nailing, sewing, weaving) offers actionable insights for workplace improvements.
Weaknesses and Suggestions
- Sample Size and Demographics:The small sample size (n=20) limits generalizability. Consider collaborating with other Krajood communities or extending data collection over a longer period. All participants were female. If males are involved in related tasks (e.g., agricultural roles mentioned), discuss how their exclusion might affect the results.
- RULA Limitations: Acknowledge that RULA primarily assesses static postures. Clarify how dynamic movements (e.g., repetitive hammering) were analyzed using video recordings. Consider supplementing RULA with complementary tools (e.g., REBA for lower-body assessment) in future work.
- Study Design: The cross-sectional design precludes causal inferences. Recommend longitudinal studies to establish causality between postures and MSDs.
- Data Presentation: Table 4 (RULA scores by worker/stage) appears incomplete in the manuscript. Ensure all data are clearly labeled and fully represented. Include visual aids (e.g., ergonomic risk heatmaps or posture diagrams) to enhance clarity.
- Discussion and Comparisons: Compare findings with similar studies in other handicraft sectors (e.g., weaving, sewing) to contextualize the uniqueness of Krajood work risks. Expand on specific ergonomic interventions (e.g., adjustable seating, tool redesign) for high-risk stages like nailing.
- Conclusion: Provide actionable recommendations (e.g., workstation redesign, rest breaks, training) rather than general statements about "immediate improvements."
The manuscript is generally comprehensible but requires significant language improvements to meet the standards of academic writing. Frequent grammatical errors, including incorrect verb tenses, subject-verb agreement, and article usage.
-
Example: "Data was collected" → "Data were collected."
-
Example: "using Krajood as main raw material" → "using Krajood as the main raw material."
Some sentences are overly complex or awkwardly constructed, hindering readability.
-
Example: "The scores for the lower back reached 100% coverage" → "The lower back exhibited a 100% prevalence of symptoms."
Missing or misplaced commas, particularly in compound sentences and after introductory phrases.
-
Example: "In 2019 globally 2.41 billion individuals..." → "In 2019, globally, 2.41 billion individuals..."
Author Response
3.Comments and Suggestions for Authors
This study provides a meaningful contribution to understanding ergonomic risks among traditional Krajood handicraft workers in Southern Thailand. The focus on musculoskeletal disorders (MSDs) in an under-researched occupational group is commendable. Below are key strengths, weaknesses, and suggestions for improvement:
Strengths
Relevance: Addresses a critical occupational health issue in a traditional craft, highlighting the need for ergonomic interventions in artisanal sectors.
Methodology: Appropriate use of RULA and Nordic Musculoskeletal Questionnaire (NMQ), combined with video analysis, strengthens the validity of findings.
Practical Implications: Clear identification of high-risk work stages (nailing, sewing, weaving) offers actionable insights for workplace improvements.
Weaknesses and Suggestions
Sample Size and Demographics:The small sample size (n=20) limits generalizability. Consider collaborating with other Krajood communities or extending data collection over a longer period. All participants were female. If males are involved in related tasks (e.g., agricultural roles mentioned), discuss how their exclusion might affect the results.
-This study is a very special occupation that limited males are interested in. However, we have discuss about the gender in a revised manuscript.
RULA Limitations: Acknowledge that RULA primarily assesses static postures. Clarify how dynamic movements (e.g., repetitive hammering) were analyzed using video recordings. Consider supplementing RULA with complementary tools (e.g., REBA for lower-body assessment) in future work.
-This statement was provided in the revised manuscript.
Study Design: The cross-sectional design precludes causal inferences. Recommend longitudinal studies to establish causality between postures and MSDs.
Data Presentation: Table 4 (RULA scores by worker/stage) appears incomplete in the manuscript. Ensure all data are clearly labeled and fully represented. Include visual aids (e.g., ergonomic risk heatmaps or posture diagrams) to enhance clarity.
-I added more information in the new Table 5.
Discussion and Comparisons: Compare findings with similar studies in other handicraft sectors (e.g., weaving, sewing) to contextualize the uniqueness of Krajood work risks. Expand on specific ergonomic interventions (e.g., adjustable seating, tool redesign) for high-risk stages like nailing.
-The manuscript was revised.
Conclusion: Provide actionable recommendations (e.g., workstation redesign, rest breaks, training) rather than general statements about "immediate improvements."
-The manuscript was added more information in the conclusion as the comments.
The manuscript is generally comprehensible but requires significant language improvements to meet the standards of academic writing. Frequent grammatical errors, including incorrect verb tenses, subject-verb agreement, and article usage.
- Example: "Data was collected" → "Data were collected."
- Example: "using Krajood as main raw material" → "using Krajood as the main raw material."
Some sentences are overly complex or awkwardly constructed, hindering readability.
- Example: "The scores for the lower back reached 100% coverage" → "The lower back exhibited a 100% prevalence of symptoms."
Missing or misplaced commas, particularly in compound sentences and after introductory phrases.
- Example: "In 2019 globally 2.41 billion individuals..." → "In 2019, globally, 2.41 billion individuals..."
- The manuscript was proofread and edited for grammatical errors by a professional academic writer. However, If you require more accuracy, we may use MDPI services.
Round 2
Reviewer 1 Report
Comments and Suggestions for Authors
Dear Authors,
You have revised your manuscript in accordance with the comments. Thank you for your effort and careful response to each comment during the review process.
I consider the final version to be competent for publication in the journal.
Reviewer 2 Report
Comments and Suggestions for Authors
I have reviewed the revised article. I have no further comments.